# Cytoplasmic translocation of the retinoblastoma protein disrupts sarcomeric organization

**Keigo Araki[1]\*[†], Keiko Kawauchi[2], Hiroaki Hirata[2], Mie Yamamoto[3], Yoichi Taya[1,4]\***

[1]Cancer Science Institute of Singapore, National University of Singapore, Singapore, Singapore; [2]Mechanobiology Institute, National University of Singapore, Singapore, Singapore; [3]Department of Pharmacology, National University of Singapore, Singapore, Singapore; [4]Department of Biochemistry, National University of Singapore, Singapore, Singapore

**Abstract** Skeletal muscle degeneration is a complication arising from a variety of chronic diseases including advanced cancer. Pro-inflammatory cytokine TNF-α plays a pivotal role in mediating cancer-related skeletal muscle degeneration. Here, we show a novel function for retinoblastoma protein (Rb), where Rb causes sarcomeric disorganization. In human skeletal muscle myotubes (HSMMs), up-regulation of cyclin-dependent kinase 4 (CDK4) and concomitant phosphorylation of Rb was induced by TNF-α treatment, resulting in the translocation of phosphorylated Rb to the cytoplasm. Moreover, induced expression of the nuclear exporting signal (NES)-fused form of Rb caused disruption of sarcomeric organization. We identified mammalian diaphanous-related formin 1 (mDia1), a potent actin nucleation factor, as a binding partner of cytoplasmic Rb and found that mDia1 helps maintain the structural integrity of the sarcomere. These results reveal a novel non-nuclear function for Rb and suggest a potential mechanism of TNF-α-induced disruption of sarcomeric organization.

**\*For correspondence:**
keigoaraki.res@gmail.com (KA);
yoichitaya99@gmail.com (YT)

[†]**Present address:**
Mechanobiology Institute, National University of Singapore, Singapore, Singapore

**Competing interests:** The authors declare that no competing interests exist.

**Reviewing editor**: Giulio Cossu, University College London, United Kingdom

## Introduction

Skeletal muscle degeneration, which is characterized by the progressive depletion of muscle strength, occurs in a variety of chronic diseases including advanced cancer, congestive heart failure, and AIDS (*Tisdale, 2002*). The underlying intercellular mechanism is currently thought to be multifactorial. Inflammatory cytokines, particularly TNF-α, have been shown to be key mediators of cancer-related skeletal muscle degeneration (*Tisdale, 2002; Seruga et al., 2008*). Elevated levels of TNF-α precede the onset of cancer-related skeletal muscle degeneration and act through several cancer-related signaling pathways such as the p53 and nuclear factor kappa B (NF-κB) pathways (*Guttridge et al., 2000; Cai et al., 2004; Schwarzkopf et al., 2006*).

Retinoblastoma protein (Rb) prevents tumor formation by inducing differentiation, controlling cell-cycle progression, and maintaining genomic stability (*Burkhart and Sage, 2008*). To date, numerous studies of Rb function have focused on the transcriptional regulation of E2F. Rb forms a transcriptional repressor complex with two protein groups, E2F transcription factors and LXCXE motif-containing proteins (*Halaban, 2005; Burkhart and Sage, 2008*). Rb activity is regulated by sequential phosphorylation on several serine and threonine residues, first by cyclin D/cyclin-dependent kinase 4 (CDK4) and then by cyclin E/CDK2 complexes (*Halaban, 2005*). This serial phosphorylation of Rb induces dissociation of the transcriptional repressor complex, allowing expression of E2F-target genes, which are required for many cellular processes. Loss of Rb function in many cancer cells is frequently caused by aberrant CDK-mediated phosphorylation (*Chau and Wang, 2003; Burkhart and Sage, 2008*). Consequently, selective CDK inhibition is considered a potentially useful approach for cancer treatment (*Malumbres*

**eLife digest** Skeletal muscles, such as the biceps and calves, are one of three main muscle groups in the body, and a range of chronic diseases—including cancer, heart disease and AIDS—can cause wasting and a loss of strength in these muscles. Many different cellular processes are known to be involved in the degeneration of skeletal muscle during illness.

For example, in people suffering from cancer, the immune response produces large numbers of molecules called inflammatory cytokines to combat the cancer cells, and these molecules are thought to have a role in the breakdown of skeletal muscle. A cytokine called tumour necrosis factor alpha, or TNF-α for short, is thought to cause muscle damage, but the details of this process are not fully understood.

One possibility is that TNF-α interacts with a protein called Rb—short for retinoblastoma protein—that suppresses the proliferation of cells that leads to cancer. However, if this protein is modified by a chemical process called phosphorylation, the Rb molecules will not be able to suppress the genes that lead to excessive cell growth. The hyperphosphorylation of Rb has been observed in many cancer cells, and it has been shown that high levels of TNF-α in cells results in Rb not working properly, but it has not been clear if faulty Rb also leads to the breakdown of skeletal muscle.

Now Araki et al. provide evidence that the phosphorylation of Rb by TNF-α leads to skeletal muscle degeneration. Araki et al. found that in muscle cells that contain high concentrations of TNF-α, the Rb molecules move from the nuclei of the cells, where they interact with genes, to the cytoplasm, where they disrupt the formation of structural fibres. This means that Rb inhibits the ability of muscle cells to slide over one during contractions and relaxation, as happens in normal muscle tissue. If confirmed by further experiments, these results could lead to the development of new approaches for the treatment of skeletal muscle degeneration.

*and Barbacid, 2009*). In addition, inactivation of Rb, which is induced by TNF-α treatment, has been shown to lead to various cellular behaviors including proliferation of vascular smooth muscle cells (*Rastogi et al., 2012*) and apoptosis of fibroblasts and aortic endothelial cells (*Chau and Wang, 2003*; *Rastogi et al., 2012*). Recently, a non-nuclear function of Rb has been reported; where Rb at the mitochondria participates in TNF-α-induced apoptosis (*Hilgendorf et al., 2013*). However, it is still unknown whether the Rb pathway is involved in cancer-related skeletal muscle degeneration mediated by TNF-α.

A sarcomere is the basic functional unit of striated muscle and consists of two sets of filaments: thick and thin (*Squire, 1997*; *Gautel, 2011*). The thick filaments are composed of myosin proteins and the thin filaments are assembled from polymerized actin monomers, called filamentous actin (F-actin). The contractile activity of skeletal muscle is achieved through the actin and myosin filaments sliding past one another (*Squire, 1997*). The motor function of striated muscle therefore, requires the well-ordered assembly of sarcomeres, which is closely tied to the highly organized actin cytoskeleton. The formins are a large family of proteins and are characterized by the presence of the conserved formin homology 2 (FH2) domain (*Kovar, 2006*; *Campellone and Welch, 2010*). The FH2 domain promotes actin nucleation and polymerization, thereby producing long straight actin filaments and regulating cytoskeletal organization. It has been reported that several members of the formin family serve as key regulators of actin dynamics during sarcomeric organization in striated muscle (*Taniguchi et al., 2009*; *Kan et al., 2012*; *Mi-Mi et al., 2012*).

In this study, we present a potential mechanism underlying TNF-α-induced skeletal muscle degeneration. We propose a novel function for Rb; where Rb disrupts sarcomeric organization in human skeletal muscle myotubes (HSMMs) following its phosphorylation and translocation into the cytoplasm. Our study implicates the tumor suppressor protein in the regulation of cytoskeletal organization.

## Results

### Phosphorylation and cytoplasmic translocation of Rb are induced by TNF-α treatment

To gain insights into the role of Rb in cancer-related skeletal muscle degeneration, we first examined the phosphorylation kinetics and subcellular localization of Rb in TNF-α-treated HSMMs. For this purpose, cells were pretreated with interferon-gamma (IFN-γ, 100 ng/ml) for 8 hr prior to initial TNF-α

treatment in order to promote cellular sensitivity to the effects of TNF-α (*Tsujimoto et al., 1986*). During differentiation from myoblasts to myotubes (from day 0 to day 4), Rb shifted from a phosphorylated to an unphosphorylated state (*Figure 1A*). Moreover, following TNF-α treatment Rb phosphorylation on the CDK4-specific phosphorylation site, serine 780 (*Kitagawa et al., 1996*), was induced (*Figure 1A*). This was not observed on threonine 821, a CDK2-specific phosphorylation site (*Halaban, 2005*). In untreated HSMMs, Rb was primarily present in the nucleus, but translocated to the cytoplasm after TNF-α treatment (*Figure 1B,C*). In addition, we found that phosphorylated Rb in TNF-α-treated HSMMs was predominantly located in the cytoplasm (*Figure 1D*). In accordance with the induction of Rb phosphorylation on a CDK4-specific phosphorylation site, TNF-α treatment led to an increase in the level of nuclear CDK4 (*Figure 1E*). The vast majority of nuclear Rb was in an unphosphorylated state, while cytoplasmic Rb that accumulated after TNF-α treatment was in a phosphorylated state (*Figure 1E*). We then tested whether TNF-α-induced cytoplasmic accumulation of Rb is caused by CDK4-mediated Rb phosphorylation. When CDK4 was depleted by short hairpin RNA (shRNA), cytoplasmic accumulation of Rb induced by TNF-α was decreased (*Figure 1F–H*). These results suggest that phosphorylation of Rb by CDK4 triggers its cytoplasmic translocation in HSMMs.

In human skeletal muscle myoblasts, Rb was mainly in a phosphorylated state (*Figure 1A*, day 0), although the localization of S780-phosphorylated Rb was confined to the nucleus (*Figure 2A*). Lamina-associated polypeptide (LAP) 2α, a binding partner of nucleoplasmic A-type lamins, interacts with S780-phosphorylated Rb in mitotic myoblasts (*Markiewicz et al., 2005*) and is known to play an important role in nuclear tethering of Rb (*Markiewicz et al., 2002*). Indeed, when LAP2α was depleted by shRNA in human myoblasts, Rb localized to the cytoplasm as well as the nucleus (*Figure 2B*). Given the level of LAP2α decreased during muscle differentiation (*Markiewicz et al., 2005*) (*Figure 2C*), these results suggest that low-level LAP2α expression in HSMMs facilitates the cytoplasmic translocation of Rb. In many types of cancer cells, Rb exists predominantly in a phosphorylated state, but is primarily localized in the nucleus. It has been reported that *LAP2α* is an E2F-target gene and its expression is enhanced in cancer cells (*Parise et al., 2006*; *Ward et al., 2011*). Overexpressed LAP2α may therefore serve to tether Rb to the nucleus in cancer cells and the cytoplasmic translocation of Rb may be triggered in cells that express low levels of LAP2α, such as terminally differentiated cells.

## Sarcomeric organization of HSMMs is impaired by TNF-α treatment

We next tested whether the motor function of skeletal muscle is affected under this condition. In response to electric pulse stimulation (EPS), which evokes a contractile reaction in myotubes (*Fujita et al., 2007*), approximately 55% of HSMMs displayed beating, an index of contractile reaction (*Figure 3A*; *Video 1*), while it was hardly observed after TNF-α treatment (*Figure 3A*; *Video 2*). The beating was not induced in mitotic myoblasts or prematurely differentiated myoblasts implying that the sarcomeric structure, which is essential for the contractile activity of muscle cells (*Squire, 1997*), is organized in HSMMs. To evaluate the effect of TNF-α on the periodic assembly of sarcomeres, we examined the distribution of α-actinin, a major component of Z-disks, as Z-disks define the lateral borders of individual sarcomeres (*Gautel, 2011*). In untreated HSMMs, α-actinin was observed at evenly spaced intervals, whereas in TNF-α-treated HSMMs the peak-to-peak distance in line plots of α-actinin intensity along the myofibril was larger and inconsistent (*Figure 3B–D*). Furthermore, when the repeating pattern of α-actinin distribution was analyzed using an autocorrelation image processing technique, peak values were much lower in TNF-α-treated HSMMs (*Figure 3E*). These results indicate that the periodic assembly of sarcomeres is disrupted by TNF-α treatment.

In striated muscle cells, anti-parallel actin filaments spanning the sarcomeres are crosslinked to the Z-disks (*Sparrow and Schock, 2009*). The peak-to-peak distance in line plots of α-actinin intensity therefore reflects a counterbalance between tensile forces generated by the actin filaments. Sarcomeric disorganization represents an imbalance in the tensile forces, which may be caused by defective actin filament formation. The close association of accurate sarcomeric organization with actin polymerization was demonstrated by treatment of the cells with the actin polymerization inhibitor cytochalasin D. After 30 min of treatment, the periodic arrangement of α-actinin was not well ordered (*Figure 4A–C*) and was strongly disordered by 60 min (*Figure 4D*).

## Cytoplasmic Rb disorganizes sarcomeric assembly

We then evaluated the role of Rb in the negative regulation of sarcomeric organization using Rb knockdown HSMMs (*Figure 5A,B*). Rb is known to be required for myogenic differentiation but not for the

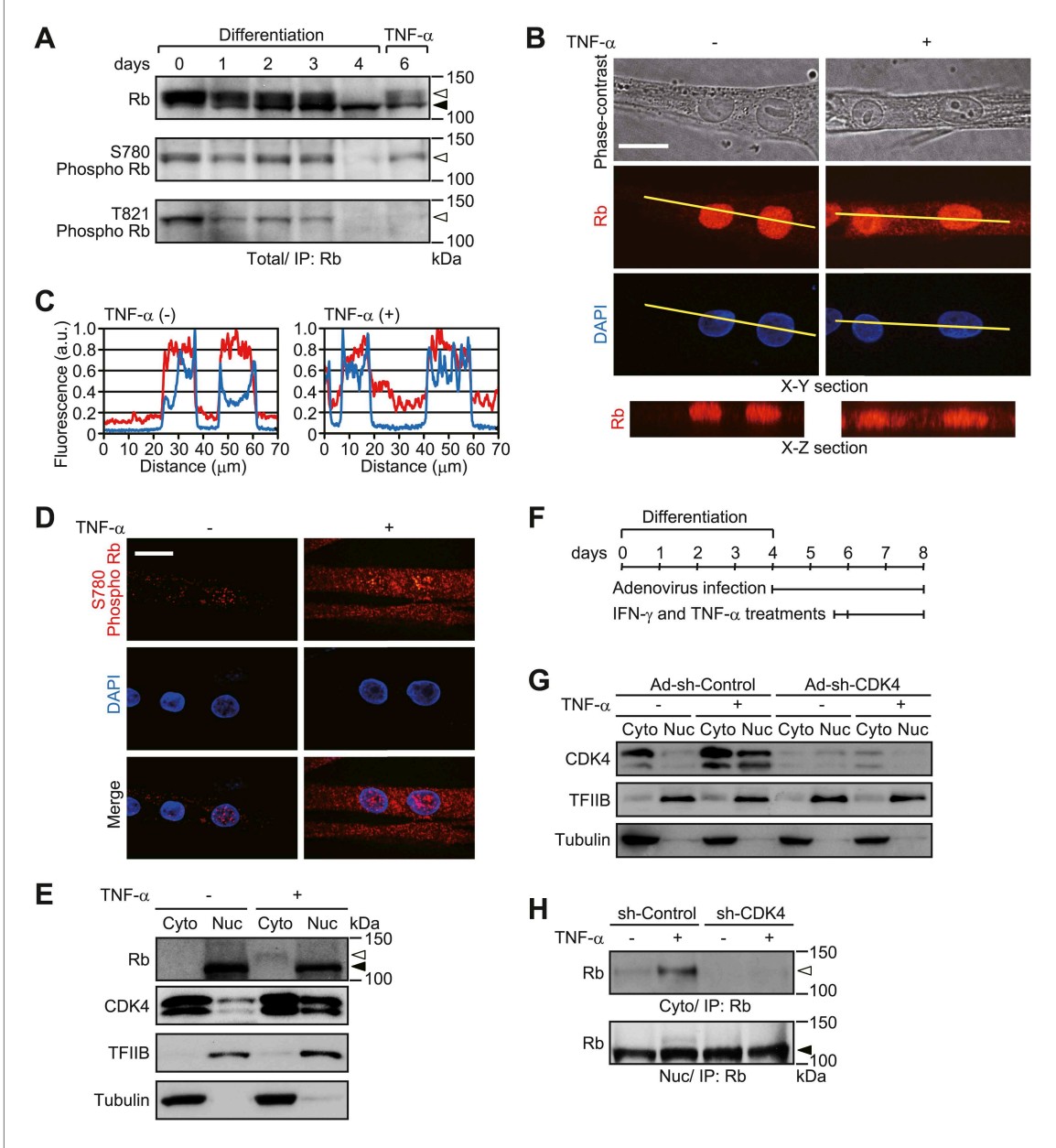

**Figure 1**. TNF-α induces cytoplasmic translocation of Rb. (**A–E**) HSMMs were treated with TNF-α for 2 days. (**A**) CDK4-mediated phosphorylation of Rb is induced by TNF-α treatment. At the indicated time points after differentiation stimuli, total cell lysates were prepared and immunoprecipitated with anti-Rb antibody, followed by immunoblotting. TNF-α was added for the last 2 days of 6-day cultures. The open and solid arrowheads indicate the position of phosphorylated and unphosphorylated Rb, respectively. (**B**) Cytoplasmic translocation of Rb is caused by TNF-α treatment. Z-stack confocal images of Rb and 4′,6-diamidino-2-phenylindole (DAPI)-stained nuclei were obtained (50 slices at 0.3-μm intervals). X-Y section images for Rb and DAPI-stained nuclei and X-Z section images for Rb along yellow lines in the X-Y section images are shown. Scale bar, 20 μm. (**C**) Line plots denote the fluorescence intensities of Rb (red lines) and DAPI (blue lines) along the yellow lines in **B**. Intensity values were normalized by the maximum value of each plot. a.u., arbitrary units. (**D**) Phosphorylated Rb is localized in the cytoplasm. Confocal images for Rb phosphorylated at S780 (red) and DAPI-stained nuclei (blue). Scale bar, 20 μm. (**E**) Nuclear CDK4 expression is increased after TNF-α treatment. Cytoplasmic (Cyto) and nuclear (Nuc) lysates were analyzed by immunoblotting with the antibodies indicated. Tubulin and TFIIB were used as loading controls for cytoplasmic and nuclear lysates, respectively. The open and solid arrowheads indicate the position of phosphorylated and unphosphorylated Rb. (**F–H**) HSMMs were infected with adenoviruses expressing control non-target shRNA or shRNA against *CDK4* at a MOI of 10 pfu/nucleus and then treated with TNF-α for 2 days. (**F**) Experimental design and reference time frame. (**G**) The cytoplasmic (Cyto) and nuclear (Nuc) lysates were analyzed by immunoblotting. (**H**) TNF-α-induced cytoplasmic translocation of Rb is prevented by CDK4 depletion. The cytoplasmic and nuclear lysates were subjected to immunoprecipitation and probed by immunoblotting. The open and solid arrowheads indicate the position of phosphorylated and unphosphorylated Rb, respectively.

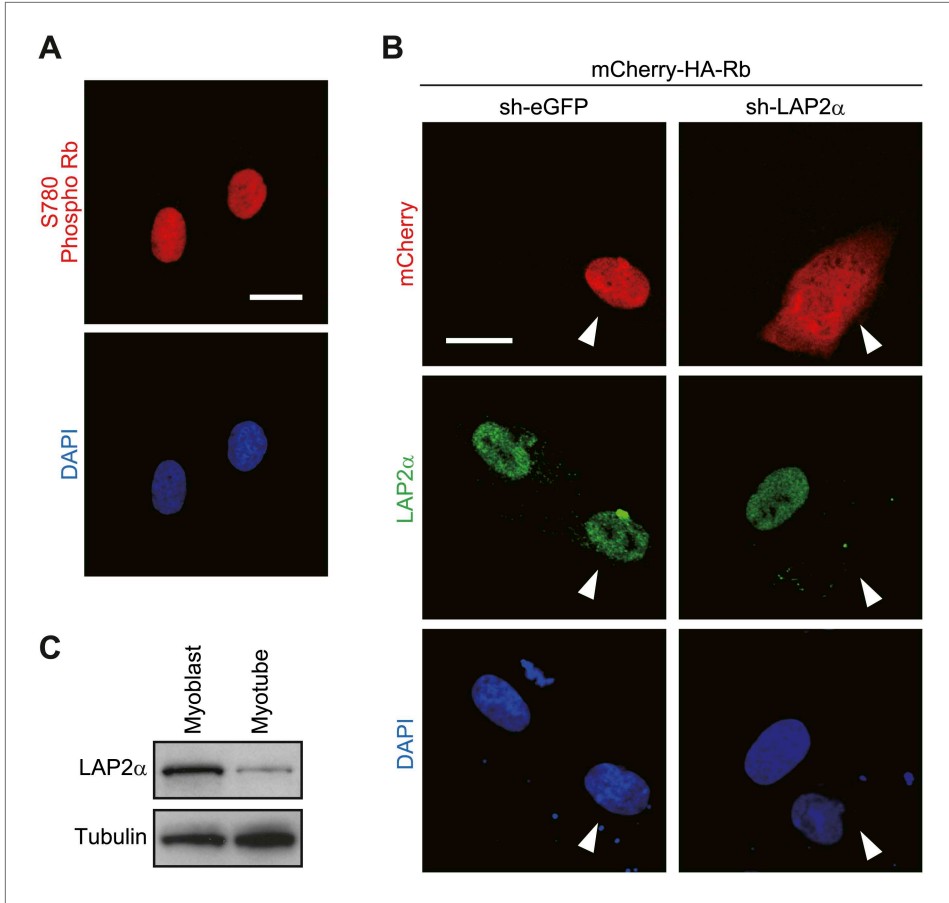

**Figure 2**. Loss of LAP2α affects the cytoplasmic translocation of Rb. (**A**) Phosphorylated Rb is localized in the nucleus in human skeletal muscle myoblasts. Confocal images for Rb phosphorylated at S780 and DAPI-stained nuclei. Scale bar, 20 μm. (**B**) Rb is translocated to the cytoplasm by LAP2α depletion. Human skeletal muscle myoblasts were transfected with an mCherry-HA-Rb expression plasmid together with an shRNA expression plasmid against *eGFP* or *LAP2α*. Confocal images for mCherry, LAP2α and DAPI-stained nuclei. The arrowheads indicate transfected cells. Scale bar, 20 μm. (**C**) LAP2α expression is decreased in HSMMs. Total cell lysates from human skeletal muscle myoblasts and HSMMs were subjected to immunoblotting.

maintenance of the terminally differentiated state in myotubes (*Huh et al., 2004*). In accordance with this, sarcomeric organization was not affected when Rb was depleted by shRNA (*Figure 5C*). After TNF-α treatment, the periodic assembly of sarcomeres was disrupted in control shRNA-expressing HSMMs, but TNF-α failed to induce the sarcomeric disorganization in sh-Rb-expressing HSMMs (*Figure 5D–F*). Next, we directly examined whether cytoplasmic Rb is capable of disrupting sarcomeric organization. To this end, HSMMs were infected with adenoviruses expressing a heterologous nuclear exporting signal (NES)-fused form of Rb, which was tagged with monomeric red fluorescent protein mCherry and influenza hemaglutinin (HA) at its amino-terminus (mCherry-HA-NES Rb). Given a small population of endogenous Rb was localized in the cytoplasm after TNF-α treatment, we expressed NES Rb at a comparable level to TNF-α-induced cytoplasmic Rb (*Figure 6A*, asterisk vs the open arrowhead). Under this condition, sarcomeric organization was not well ordered as in TNF-α-treated HSMMs (*Figure 6B–D*). When we expressed higher levels of NES Rb in HSMMs (*Figure 6E,F*), the periodic arrangement of α-actinin was significantly disordered when compared to mCherry-HA-Rb-expressing cells (*Figure 6G*). Taken together, these results suggest that Rb is involved in TNF-α-induced sarcomeric disorganization and TNF-α-induced cytoplasmic Rb may have a pivotal role in this process.

Similarly to TNF-α-treated HSMMs, the cytoplasmic translocation of Rb has been shown to be induced by its CDK-mediated phosphorylation in certain types of cancer cells (*Jiao et al., 2008*), although

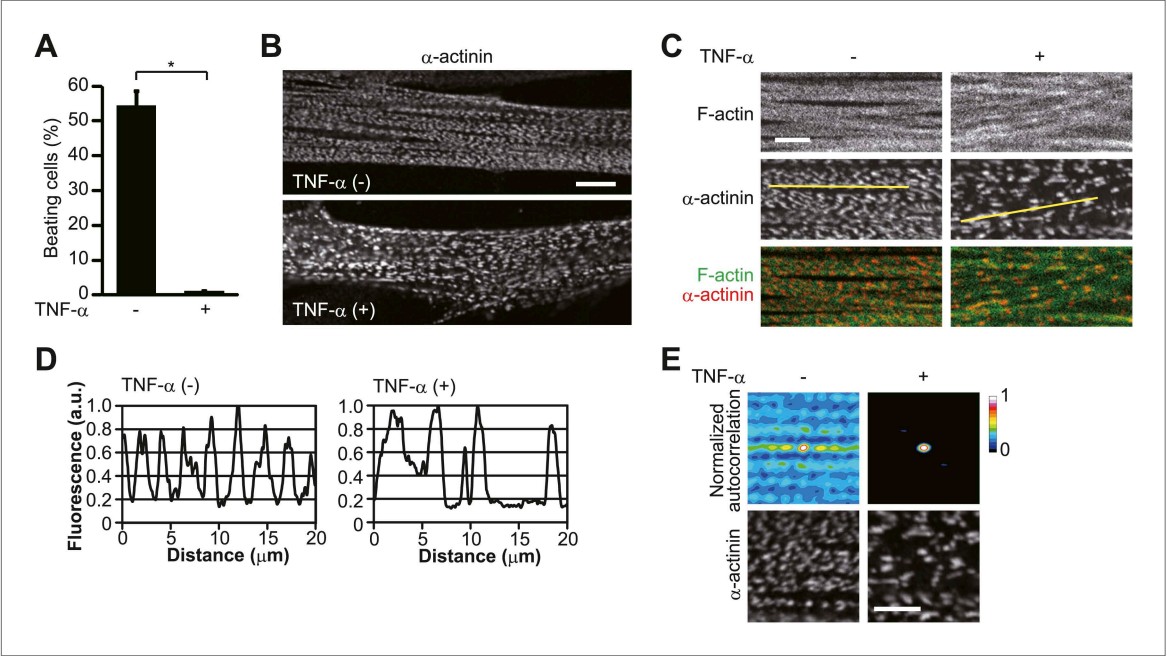

**Figure 3**. TNF-α disrupts sarcomeric organization. (**A–E**) HSMMs were treated with TNF-α for 2 days. (**A**) Contractile activity of HSMMs is impaired by TNF-α treatment. EPS was applied to HSMMs. The percentage of beating cells from a total of 100 HSMMs is shown. Results are presented as mean ± SD from three independent experiments. *p<0.002, determined by the Student's *t*-test. (**B–E**) Sarcomeric organization of HSMMs. Confocal images for α-actinin (**B**) and merged images of F-actin (green) and α-actinin (red) (**C**) are shown. Scale bar, 10 μm in **B** and 5 μm in **C**. (**D**) Line plots of α-actinin fluorescence intensity along individual myofibrils (denoted by yellow lines in **C**). Intensity values were normalized by the maximum value for each fibril. a.u., arbitrary units. (**E**) Autocorrelation analyses of the α-actinin distribution. Scale bar, 5 μm.

the function of cytoplasmic Rb in the cancer cells has not been described thus far. In the cancer cells, Rb is phosphorylated on CDK2 phosphorylation sites as well as CDK4 phosphorylation sites, whereas our data showed that Rb phosphorylation on the CDK2-specific phosphorylation site, threonine 821, was not induced by TNF-α treatment (*Figure 1A*, day 6). We therefore reasoned that the selective phosphorylation of Rb by specific CDKs affects the function of cytoplasmic Rb. Given that it has been reported that threonine 821/threonine 826 phosphorylation disrupts Rb binding to LXCXE motif-containing proteins (*Knudsen and Wang, 1996*; *Dick and Rubin, 2013*), we examined whether the function of cytoplasmic Rb is achieved through its interaction with LXCXE motif-containing proteins. We introduced mCherry-HA-NES Rb lacking exon 22 (Rb Δexon 22), which is known to be an LXCXE-binding deficient mutant (*Henley et al., 2010*), into HSMMs and found that the inhibitory effect of NES Rb Δexon 22 on sarcomeric organization was less effective as compared to NES Rb-expressing HSMMs (*Figure 7A*).

Next, to explore how cytoplasmic Rb disorganizes sarcomeric assembly in HSMMs, we searched for the binding proteins of cytoplasmic Rb using mCherry-HA-NES Rb wild-type (WT)-expressing HSMMs. Total cell lysates from adenovirus-infected HSMMs were immunoprecipitated with anti-HA antibody-conjugated agarose beads

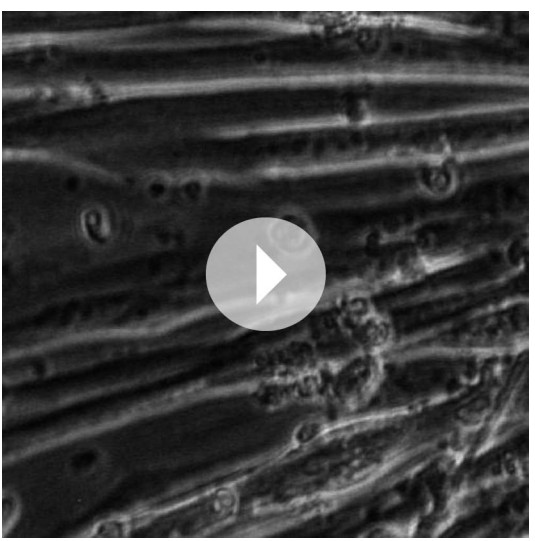

**Video 1**. Image of live beating HSMMs in response to EPS.

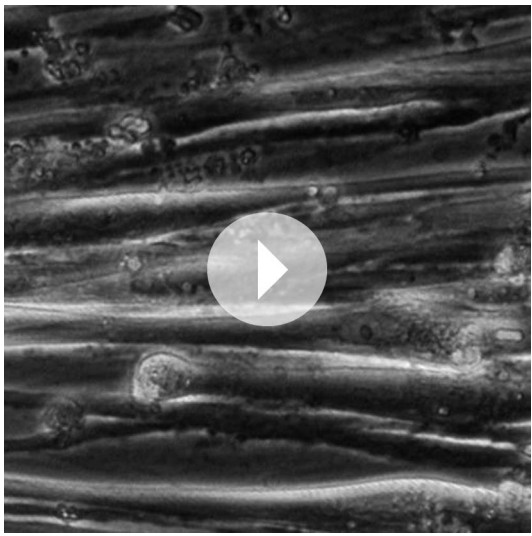

**Video 2**. Image of live beating TNF-α-treated HSMMs in response to EPS.

and the bound proteins were subjected to electro-spray ionization mass spectrometric analysis (*Figure 7—figure supplement 1*). Among the proteins identified, we focused on mammalian diaphanous-related formin 1 (mDia1), a potent actin nucleation factor (*Watanabe et al., 1997*), as the LXCXE motif is contained in the GTPase binding domain (GBD) of mDia1 (*Figure 7B,C*). It is noteworthy that the band including mDia1 was hardly detected in mCherry-HA-NES Rb Δexon 22-expressing HSMMs (*Figure 7—figure supplement 1*, band 1). Data from purified wild-type and LXCXE motif-deleted mutant mDia1 proteins (WT and ΔLXCXE) (*Figure 7C*) showed that binding of mDia1 to Rb was abolished by deletion of the LXCXE motif (*Figure 7D*). Cytoplasmic Rb, which was phosphorylated and accumulated after TNF-α treatment, interacted with mDia1 (*Figure 7E*). We then examined whether the selective phosphorylation of Rb affects this inter-action. Unphosphorylated Rb protein containing the large pocket, an important domain for inter-

actions with a variety of cellular proteins (*Burkhart and Sage, 2008*), bound to mDia1 in vitro and the binding level was strongly reduced according to its phosphorylation catalyzed by cyclin D/CDK4 and cyclin E/CDK2 complexes (*Figure 7F,G*). Although differences in the patterns of Rb phosphoryla-tion by cyclin D/CDK4 and cyclin E/CDK2 have been reported, we failed to detect the selective phosphorylation of Rb by our in vitro phosphorylation system, which may be due to a supraphysio-logical activity of CDKs in vitro. We then tested a mutant Rb protein bearing serine/threonine-to-alanine substitutions in the CDK2-specific phosphorylation sites, serine 612 and threonine 821 (Mut CDK2). The mutant Rb exhibited substantial interaction with mDia1 even after phosphorylation (*Figure 7G*), suggesting that phosphorylation of Rb on CDK4 phosphorylation sites alone does not impair its interaction with mDia1 and TNF-α-induced cytoplasmic Rb has the potential to interact with mDia1.

Muscle wasting/atrophy accompanies cancer-related skeletal muscle degeneration (*Tisdale, 2002*; *Acharyya et al., 2004*). We therefore carried out immunohistological analysis on normal and atrophied tibialis anterior muscles excised from cancer patients and examined the localization of Rb in these muscles (*Figure 8A*). In normal muscles, Z-disks were regularly aligned and the sarco-mere striation pattern was clearly observed (*Figure 8B*, left). In contrast, the Z-disks were misaligned and the sarcomeric banding pattern was not well organized in the atrophied muscles (*Figure 8B*, right). With regard to the localization of Rb, it was mainly located in the nucleus in normal muscle cells, but Rb could be observed in the cytoplasm, as well as the nucleus in atrophied muscle cells (*Figure 8C*). In both normal and atrophied skeletal muscles, the localization of mDia1 was mainly confined to the Z-disk (*Figure 8B*), and cytoplasmic Rb observed in atrophied muscles colocalized with mDia1 (*Figure 8C*).

## TNF-α-induced damage of HSMMs is recovered by constitutively active mDia1

We next tested whether inhibition of mDia1 activity is responsible for the TNF-α-induced disorganization of the sarcomere. For this purpose, we generated a recombinant adenovirus expressing the constitu-tively active form of mDia1 (*Watanabe et al., 1999*), which lacks the entire GBD and carboxy-terminal diaphanous autoregulatory domain (DAD) (*Figure 7C*). We introduced mDia1 ΔGBD/ΔDAD tagged with green fluorescent protein (GFP-mDia1 ΔGBD/ΔDAD) into HSMMs and treated them with TNF-α (*Figure 9A*). mDia1 ΔGBD/ΔDAD itself did not affect either α-actinin distribution or the contractile reaction of HSMMs (*Figure 9B,C*). After TNF-α treatment, however, the lateral alignment of α-actinin in GFP-mDia1 ΔGBD/ΔDAD-expressing HSMMs was well ordered as compared to that in control GFP-expressing HSMMs (*Figure 9D–F*). Accordingly, the percentage of beating cells decreased by TNF-α

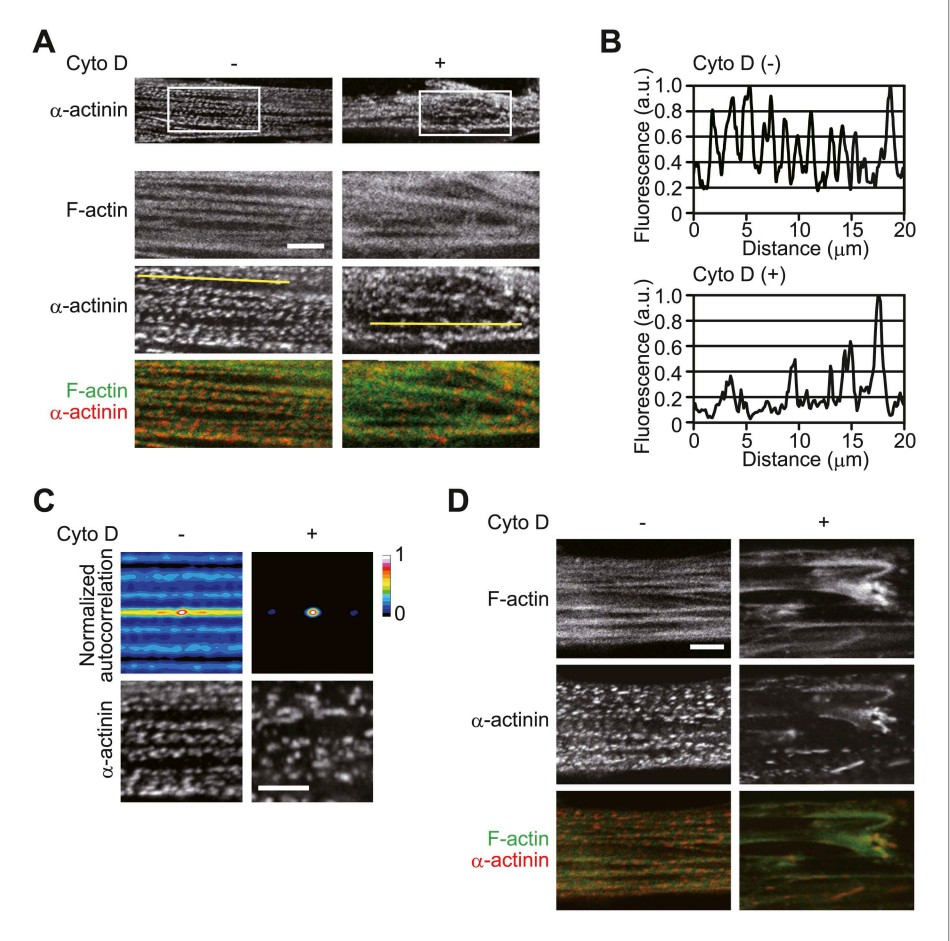

**Figure 4**. Inhibition of actin polymerization disorganizes sarcomeric assembly. (**A–D**) HSMMs were treated with 2 μM cytochalasin D (Cyto D) for 30 min (**A–C**) or 60 min (**D**) at room temperature. (**A**) The periodic arrangement of α-actinin is not well ordered after 30 min of Cyto D-treatment. Merged images of F-actin (green) and α-actinin (red). Scale bar, 5 μm. (**B**) Line plots of α-actinin fluorescence intensity along individual myofibrils (denoted by yellow lines in **A**). Intensity values were normalized by the maximum value for each fibril. a.u., arbitrary units. (**C**) Autocorrelation analyses of the α-actinin distribution. Scale bar, 5 μm. (**D**) The lateral periodicity in the α-actinin distribution is strongly disordered by 60 min of treatment. Merged images of F-actin (green) and α-actinin (red). Scale bar, 5 μm.

treatment was restored by the introduction of mDia1 ΔGBD/ΔDAD (p<0.02, determined by the Student's *t*-test) (*Figure 9C*).

## Discussion

In this study, we have elucidated a novel pathway of cancer-related skeletal muscle degeneration. TNF-α induces CDK4 activation and the concomitant phosphorylation of Rb. Subsequently, cytoplasmic translocation of Rb is triggered. Cytoplasmic Rb disrupts sarcomeric organization, which may be caused by dysfunction of mDia1. The precise role of mDia1 in the regulation of sarcomeric organization is poorly understood (*Sparrow and Schock, 2009*), but the contribution of mDia1 to sarcomeric organization is supported. When mDia1 was depleted by shRNA, the lateral periodicity in the distribution of α-actinin was markedly perturbed (*Figure 10A,B*). The importance of actin nucleation factors for the periodic assembly of sarcomeres is proposed by the recent observations that depletion of actin nucleation factors, such as Fhod3 and leiomodin, results in disordered α-actinin distribution in cardiomyocytes (*Chereau et al., 2008*; *Taniguchi et al., 2009*; *Iskratsch et al., 2010*). The notion that the degree of sarcomeric disorganization is dependent on the inhibition of actin polymerization is further supported

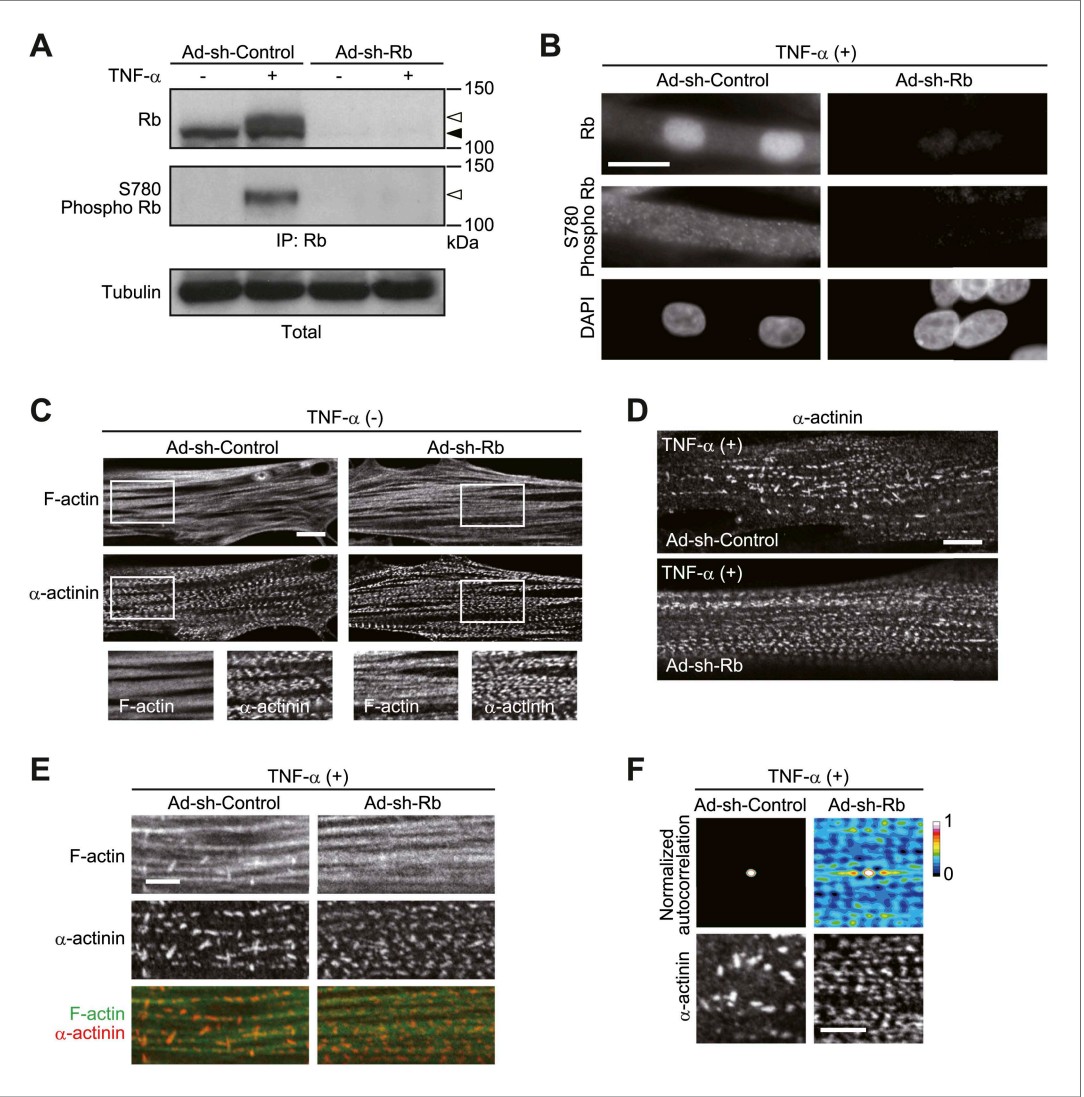

**Figure 5**. Rb contributes to TNF-α-induced sarcomeric disorganization. (**A–F**) HSMMs were infected with adenoviruses expressing control non-target shRNA or shRNA against *Rb* at a MOI of 10 pfu/nucleus and then treated with TNF-α for 2 days. (**A** and **B**) Depletion of Rb protein was verified by immunoblotting (**A**) and epifluorescence microscopy (**B**). The open and solid arrowheads indicate the position of phosphorylated and unphosphorylated Rb, respectively. Scale bar, 20 µm. (**C**) Confocal images for F-actin and α-actinin. Scale bar, 10 µm. (**D** and **E**) TNF-α-induced sarcomeric disorganization is attenuated by Rb depletion. Confocal images for α-actinin (**D**) and merged images of F-actin (green) and α-actinin (red) (**E**). Scale bar, 10 µm in **D** and 5 µm in **E**. (**F**) Autocorrelation analyses of the α-actinin distribution. Scale bar, 5 µm.

by our data that show Z-disk alignment is more severely impaired by longer-term treatment of cytochalasin D (*Figure 4*).

Phosphorylation of Rb and the subsequent activation of E2F transcriptional activity enhance the expression of E2F-target genes, which include cell-cycle regulators (e.g., *Tk1* and *Dhfr*). Although Rb phosphorylated at serine 780 is unable to bind to E2F1 (*Kitagawa et al., 1996*), quantitative PCR (qPCR) analysis did not reveal any statistically significant differences in the expression of *Tk1* and *Dhfr* after TNF-α treatment (*Figure 11*). During muscle differentiation, methylation of histone H3 lysine 9 and DNA methylation occur at several E2F-target gene promoters including *Tk1* and *Dhfr* (*Ait-Si-Ali et al., 2004*; *Blanchet et al., 2011*). These epigenetic changes may trigger the permanent silencing of E2F-target gene expression and prevent E2F1 from activating their expression after terminal differentiation.

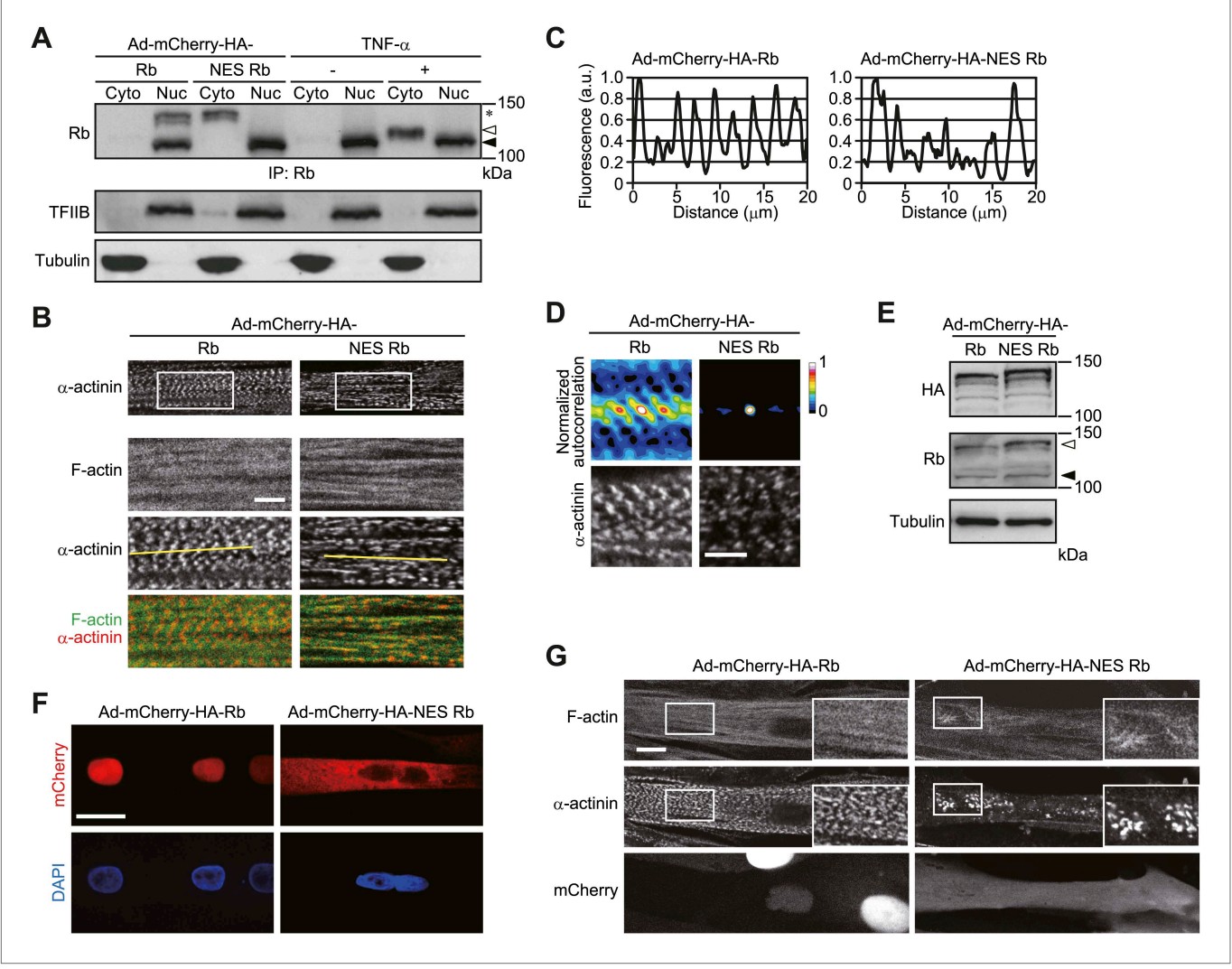

**Figure 6**. Sarcomeric organization is impaired by cytoplasmic Rb. (**A–G**) HSMMs were infected with adenoviruses expressing mCherry-HA-Rb or mCherry-HA-NES Rb at a MOI of 10 pfu/nucleus (**A–D**) or 50 pfu/nucleus (**E–G**) for 4 days. (**A**) The expression of exogenous Rb proteins. Cytoplasmic (Cyto) and nuclear (Nuc) lysates were prepared from infected HSMMs in parallel with TNF-α-treated HSMMs. The lysates were subjected to immunoprecipitation and probed by immunoblotting. The open and solid arrowheads indicate the position of phosphorylated and unphosphorylated Rb. Asterisk indicates the position of exogenous Rb. (**B**) Sarcomeric structure is not well ordered in NES Rb-expressing HSMMs. Merged images of F-actin (green) and α-actinin (red). Scale bar, 5 µm. (**C**) Line plots of α-actinin fluorescence intensity along individual myofibrils (denoted by yellow lines in **B**). Intensity values were normalized by the maximum value for each fibril. a.u., arbitrary units. (**D**) Autocorrelation analyses of the α-actinin distribution. Scale bar, 5 µm. (**E** and **F**) The expression and distribution of exogenous Rb proteins were analyzed by immunoblotting (**E**) and confocal microscopy (**F**). The open and solid arrowheads indicate the position of exogenous and endogenous Rb, respectively. Scale bar, 20 µm. (**G**) Sarcomeric structure is strongly disordered in NES Rb-expressing HSMMs. Confocal images for F-actin and α-actinin. Scale bar, 10 µm.

It has been reported that phosphorylation of Rb is induced in the skeletal muscles of mice under fasting conditions (*Blanchet et al., 2011*), which leads to the activation of E2F1 transcriptional activity and increases the expression levels of mitochondrial biogenesis factors (*Ppargc1a* and *Tfam*). In contrast, although TNF-α treatment induced Rb phosphorylation, the expression of *Ppargc1a* and *Tfam* decreased after TNF-α treatment (*Remels et al., 2010*) (*Figure 11*). These findings suggest that other TNF-α-induced signaling pathways, such as the NF-κB pathway, may affect the transcriptional activity of E2F1 (*Araki et al., 2008*) and/or regulation of the expression of mitochondrial biogenesis factors (*Remels et al., 2010*). The contractile activity of skeletal muscle is caused by the sliding of thick and thin filaments in the sarcomere, and this sliding action is intimately linked to the proper control of adenosine 5′-triphosphate (ATP) production (*Squire, 1997*). Because mitochondria are responsible for cellular

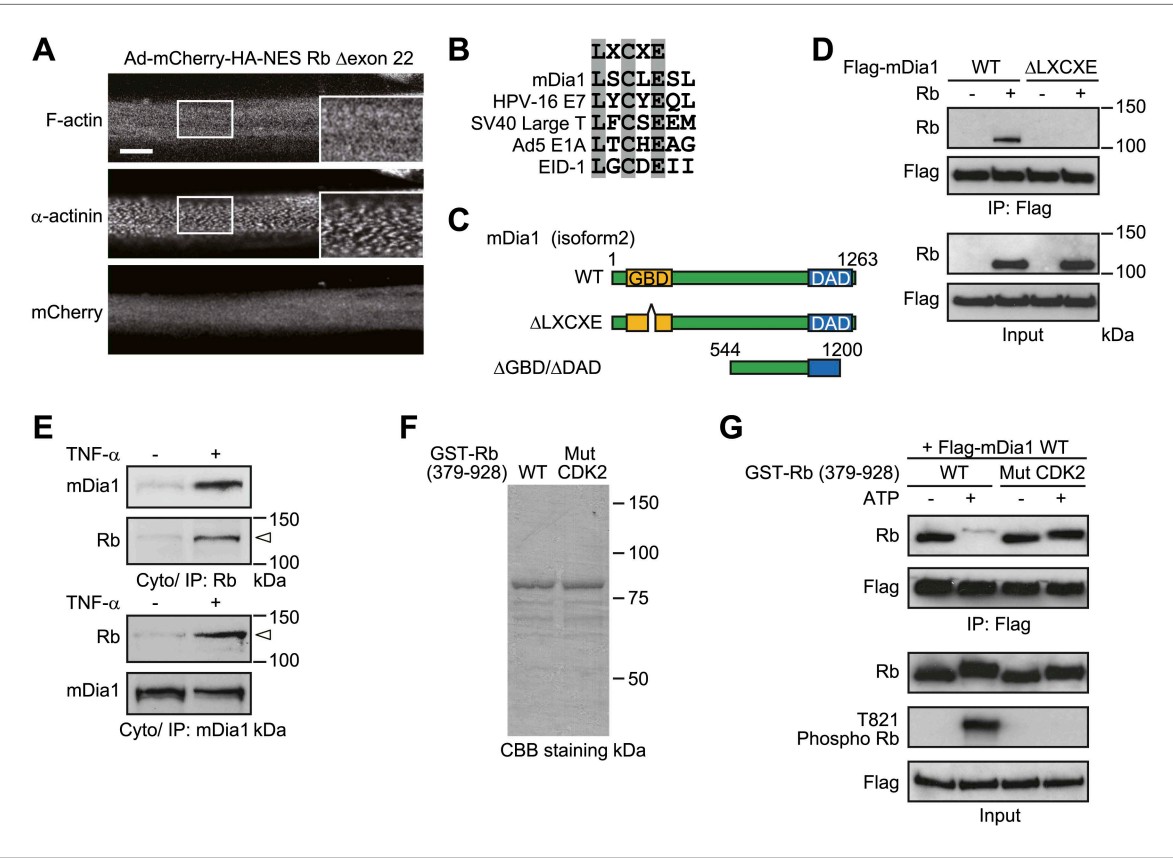

**Figure 7**. The function of cytoplasmic Rb is rendered through its interaction with LXCXE motif-containing proteins. (**A**) Sarcomeric structure is not disordered in NES Rb Δexon 22-expressing HSMMs. HSMMs were infected with an adenovirus expressing mCherry-HA-NES Rb Δexon 22 at a MOI of 50 pfu/nucleus for 4 days. Confocal images for F-actin and α-actinin. Scale bar, 10 μm. (**B**) mDia1 contains the LXCXE motif. Sequence alignment of the LXCXE motif of mDia1 and other known Rb-binding proteins. (**C**) The primary structure of mDia1 and its mutants. Scheme represents location of GBD and DAD of mDia1. Numbers denote amino acid positions in mDia1 isform2. The LXCXE motif is present in GBD (amino acid positions 153 to 157). ΔGBD/ΔDAD, doubly deleted mDia1 lacking both GBD and DAD. (**D**) The LXCXE motif is required for the in vitro interaction between Rb and mDia1. Purified Flag-tagged mDia1 proteins were mixed with full-length recombinant Rb protein and immunoprecipitates were analyzed by immunoblotting. (**E**) Rb interacts with mDia1 after TNF-α treatment. HSMMs were treated with TNF-α for 2 days. The cytoplasmic lysates were subjected to immunoprecipitation and probed by immunoblotting. The open arrowheads indicate the position of phosphorylated Rb. (**F**) Expression and purification of GST fusion Rb proteins. The purity of bacterially expressed GST-Rb proteins was evaluated by SDS-PAGE, followed by CBB staining. GST-Rb wild-type protein encompasses amino acids 379–928. GST-Rb Mut CDK2 contains serine/threonine to alanine substitutions at CDK2-specific phosphorylation sites (S612 and T821). (**G**) GST-Rb proteins were preincubated with CDK4/Cyclin D1 and CDK2/Cyclin E proteins in the presence or absence of ATP and then mixed with purified Flag-mDia1 protein. The interaction between mDia1 and Rb proteins was analyzed by immunoprecipitation with anti-Flag antibody-agarose beads.

The following figure supplements are available for figure 7:

**Figure supplement 1**. Identification of NES Rb-binding protein.

ATP production, incomplete restoration of the percentage of beating cells by mDia1 ΔGBD/ΔDAD might be ascribed to an impairment of mitochondrial function caused by TNF-α (*Figure 9C*).

In cancer patients, the circulating levels of TNF-α and IFN-γ would be lower than the amounts used in this study. Monocytes/macrophages infiltrate in atrophied muscles and may produce considerable amounts of these inflammatory cytokines. It may therefore be possible that the local concentrations of TNF-α and IFN-γ are elevated in atrophied muscles, which then contributes to phosphorylation of Rb in the long-term. TNF-α-induced skeletal muscle degeneration is achieved through multiple mechanisms. The results presented in this study propose a novel non-nuclear function for Rb, which is independent of the transcriptional regulation of E2F and may be involved in TNF-α-induced skeletal muscle degeneration. In the future, it would be interesting to clarify the role of mDia1 and determine how cytoplasmic Rb disrupts sarcomeric organization.

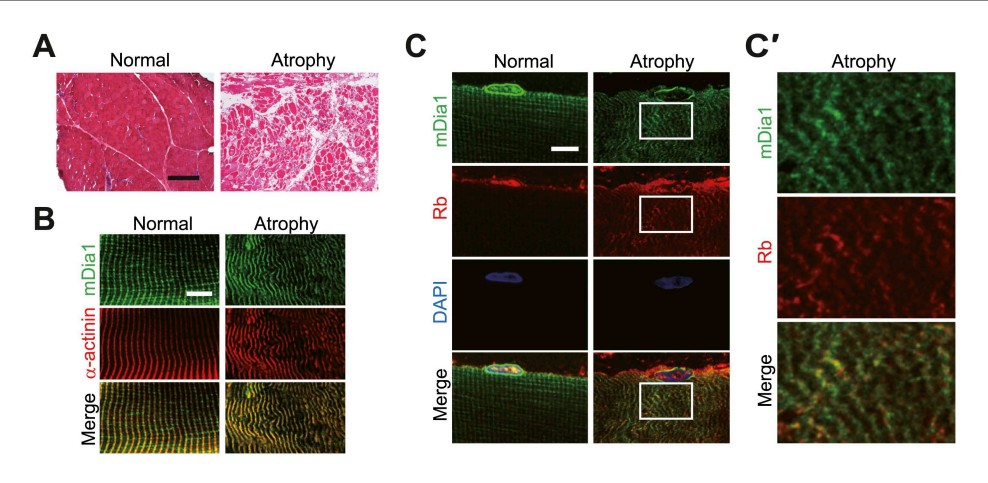

**Figure 8**. Cytoplasmic Rb colocalizes with mDia1 in atrophied skeletal muscle. (**A–C′**) Normal and atrophied tibialis anterior muscles were surgically excised from cancer patients. Cross sections (**A**) and longitudinal sections (**B–C′**) were stained. (**A**) Hematoxylin and eosin-stained cryosections. Scale bar, 400 μm. (**B–C′**) Localization of Rb and mDia1 in normal and atrophied tibialis anterior muscles. Confocal images of cryosections stained with anti-mDia1 and anti-α-actinin antibodies (**B**) or anti-Rb and anti-mDia1 antibodies (**C**, magnified in **C′**). Scale bar, 10 μm.

## Materials and methods

### Cell culture and reagents

Early passage human skeletal myoblasts purchased from Lonza (Basel, Switzerland) were cultured according to the manufacturer's instructions using Lonza-supplied growth medium (SkGM-2 BulletKit). Cells grown to approximately 80% confluence were induced to differentiate into multinucleated myotubes by switching to differentiation medium (DMEM-F12 containing 2% horse serum). Under these conditions, numerous myotubes could be detected on the fourth day. For TNF-α treatment, HSMMs were cultured in fresh serum-free media containing TNF-α (100 ng/ml) for 2 days. TNF-α was repeatedly added every 24 hr. Recombinant human TNF-α and IFN-γ were purchased from PeproTech (Rocky Hill, NJ). Cytochalasin D was obtained from Sigma (St. Louis, MO).

### Immunoprecipitation (IP) and immunoblotting

Immunoprecipitations were performed using anti-Flag antibody- (M2; Sigma) and anti-HA antibody- (3F10; Roche, Indianapolis, IN) conjugated agarose beads. The anti-Rb antibody (G3-245; BD Biosciences, San Jose, CA) and anti-mDia1 antibody (AP50, a gift from S Narumiya) (*Watanabe et al., 1997*) were used for immunoprecipitation. The following antibodies were used for immuno-blotting: anti-Rb (G3-245 and ab6075; Abcam, Cambridge, UK), anti-phospho Rb-S780 (9307; Cell Signaling Technology, Danvers, MA), anti-phospho Rb-T821 (a gift from K Tamai, CycLex), anti-CDK4 (C-22; Santa Cruz Biotechnology, Santa Cruz, CA), anti-mDia1 (AP50 and 51 [we used two kinds of mDia1 antibodies: one is AP50 which is mentioned above; the other is 51, which is a clone name of BD antibody]; BD Transduction Laboratories, Franklin Lakes, NJ), anti-LAP2α (ab5162; Abcam), anti-GFP (598; MBL, Nagoya, Japan), anti-α-Tubulin (DM1A; Sigma), anti-TFIIB (C-18; Santa Cruz Biotechnology), anti-Flag-Peroxidase (M2; Sigma) and anti-HA-Peroxidase (3F10; Roche).

### Adenovirus infections

The recombinant adenoviruses expressing mCherry-HA-Rb, mCherry-HA-NES Rb, mCherry-HA-NES Rb Δexon 22, GFP and GFP-mDia1 ΔGBD/ΔDAD were generated using the ViraPower adenoviral expression system (Invitrogen, Carlsbad, CA). The recombinant adenovirus-mCherry-HA-NES Rb contained the NES of MAPKK (NLVDLQKKLEELELDEQQ) (*Fukuda et al., 1996*) between mCherry and Rb. The recombinant adenoviruses expressing shRNAs were generated using the BLOCK-iT adenoviral RNAi

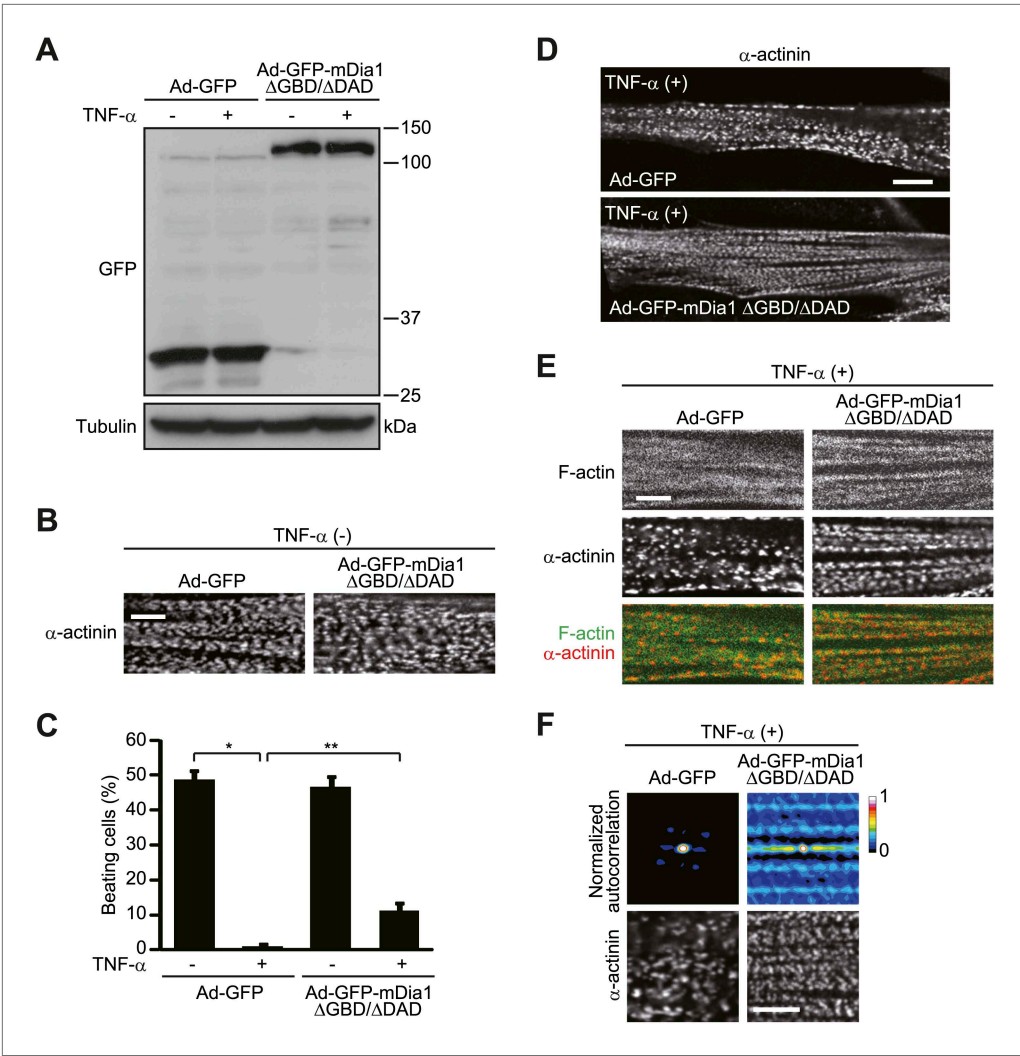

**Figure 9**. TNF-α-induced sarcomeric disorganization is prevented by constitutively active mDia1. (**A–F**) HSMMs were infected with adenoviruses expressing GFP or GFP-mDia1 ΔGBD/ΔDAD at a MOI of 1 pfu/nucleus and then treated with TNF-α for 2 days. (**A**) The expression was analyzed by immunoblotting. (**B**) Confocal images for α-actinin. Scale bar, 5 μm. (**C**) TNF-α-induced contractile dysfunction is diminished by constitutively active mDia1. EPS was applied to HSMMs. The percentage of beating cells from a total of 100 HSMMs is shown. Results are presented as mean ± SD from three independent experiments. *p<0.002; **p<0.02, determined by the Student's t-test. (**D** and **E**) Constitutively active mDia1 recovers TNF-α-induced sarcomeric disorganization. Confocal images for α-actinin (**D**) and merged images of F-actin (green) and α-actinin (red) (**E**). Scale bar, 10 μm in **D** and 5 μm in **E**. (**F**) Autocorrelation analyses of the α-actinin distribution. Scale bar, 5 μm.

expression system (Invitrogen). For shRNA-mediated gene silencing, the respective target sequences were as follows: *CDK4*, 5′-CCTAGATTTCCTTCATGCCAA-3′ (sh-CDK4); *mDia1*, 5′-GCCCAGAA TCTCTCAATCTTT-3′ (sh-mDia1); *Rb*, 5′-CAGAGATCGTGTATTGAGATT-3′ (sh-Rb); the non-target control, 5′-CAACAAGATGAAGAGCACCAA-3′ (sh-Control). The recombinant adenoviruses were purified with AsEasy virus purification kits (Agilent technologies, Palo Alto, CA) and adenovirus infections were performed with ViraDuctin adenovirus transduction reagent (Cell Biolabs, San Diego, CA).

## Plasmids

Mammalian expression vectors that encode shRNAs against *human LAP2α* or *enhanced GFP* (*eGFP*) were constructed by cloning suitable oligonucleotide sequences (*human LAP2α*, 5′-CAGAAGAGAA TTGATCAGT-3′; *eGFP*, 5′-ACAACAGCCACAACGTCTA-3′) into the pSilencer 2.1-U6 Hygro vector (Ambion, Austin, TX). pXJ Flag-mDia1 was obtained from C Koh (*Xie et al., 2008*).

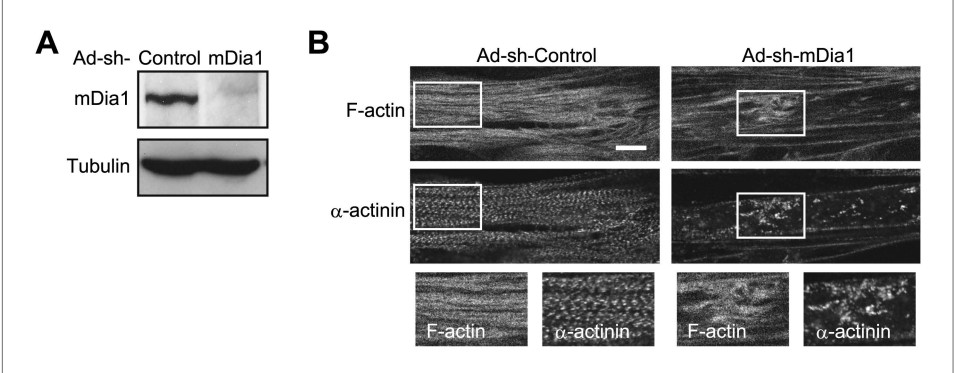

**Figure 10**. mDia1 is critical for sarcomeric organization. (**A** and **B**) HSMMs were infected with adenoviruses expressing control non-target shRNA or shRNA against *mDia1* at a MOI of 5 pfu/nucleus for 4 days. (**A**) Depletion of mDia1 protein was verified by immunoblotting. (**B**) Confocal images for F-actin and α-actinin. Scale bar, 10 µm.

## Proteins and in vitro phosphorylation of Rb

Flag-mDia1 plasmids were transfected into 293T cells and total cell lysates were extracted. The mDia1 proteins tagged with a Flag epitope at their amino-termini were captured on anti-Flag antibody-agarose beads and eluted by competition with free 1× Flag peptide (Sigma) in 50 mM Tris (pH 7.4) and 50 mM NaCl. The purity of the mDia1 proteins, wild-type and ΔLXCXE, was evaluated by SDS-polyacrylamide gel electrophoresis (SDS-PAGE), followed by Coomassie Brilliant Blue staining (CBB staining). Full-length recombinant Rb protein was purchased from QED bioscience (San Diego, CA). The recombinant glutathione *S*-transferase (GST) fusion Rb proteins were produced in BL21 *E. coli*. GST-Rb proteins were incubated with CDK4/Cyclin D1 (Merck Millipore, Billerica, MA) and CDK2/Cyclin E (Merck Millipore) proteins in the presence or absence of 100 µM ATP in the kinase buffer (20 mM Tris [pH 7.5], 10 mM MgCl$_2$ and 1 mM dithiothreitol [DTT]) for 30 min at 30°C.

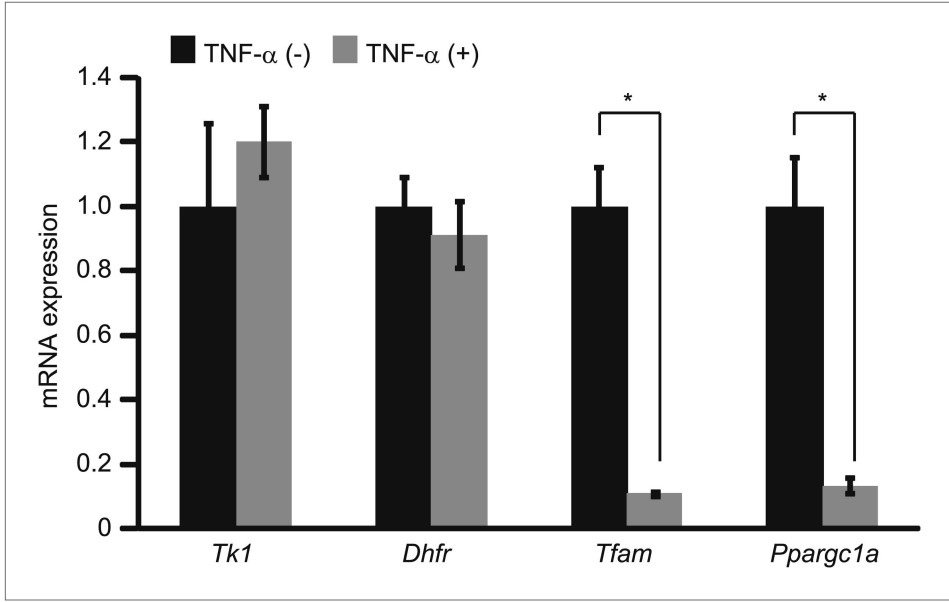

**Figure 11**. The expression of cell-cycle regulators and mitochondrial biogenesis factors after TNF-α treatment. HSMMs were treated with TNF-α for 2 days. Quantification of the expression of cell-cycle regulators and mitochondrial biogenesis factors. Results are presented as mean ± SD from three independent experiments. *p<0.02, determined by the Student's *t*-test.

## Immunofluorescence microscopy

The cells were fixed with 4% paraformaldehyde (PFA) in phosphate-buffered saline (PBS) for 30 min, permeabilized with 0.2% (vol/vol) Triton X-100/PBS for 5 min, and then blocked with 1% bovine serum albumin (BSA)/PBS for 30 min at room temperature. Subsequently, the cells were incubated with anti-Rb (1:200, 9309; Cell Signaling Technology), anti-phospho Rb-S780 (1:300, 13H9L5; Novex, Carlsbad, CA) and anti-LAP2α (1:300, ab5162; Abcam) antibodies for 1 hr, and further incubated with Alexa Fluor 546-conjugated goat anti-mouse IgG, Alexa Fluor 546-conjugated goat anti-rabbit IgG and Alexa Fluor 488-conjugated goat anti-rabbit IgG antibodies (Molecular Probes, Carlsbad, CA) for 30 min. DAPI was used for nuclear staining. For double-staining for α-actinin and F-actin, cytoskeletal stabilizing buffer was used in lieu of PBS, as described previously (*Hirata et al., 2008*). The cells were incubated with anti-α-actinin antibody (1:100, EA53; Sigma), followed by incubation with Alexa Fluor 488- or 546-conjugated goat anti-mouse IgG antibody and Alexa Fluor 546- or 633-phalloidin (Molecular Probes). Confocal images were taken using a PerkinElmer Spinning Disk microscope or a Nikon A1Rsi microscope, equipped with oil-immersion objectives (60× and 100×). Epifluorescence images were taken using a Nikon A1Rsi microscope, equipped with an oil-immersion objective (60×) and an electron multiplying charge-coupled device camera (DU897; Andor technology, Belfast, UK). Images were acquired with the Volocity software and the Nikon NIS-Elements imaging software. To evaluate periodicity in the distribution of α-actinin, 256 × 256 pixel, corresponding to 12 × 12 μm, sample areas of fluorescence images of α-actinin were subjected to computing autocorrelation analyses as described previously (*Peterson et al., 2004*) using the ImageJ program (NIH, version 10.2). Each autocorrelation image was then normalized by the value of the central peak in the image. Features of periodicity of its distribution are represented as local peaks other than the central maxima. Sample images of α-actinin and normalized autocorrelation images are shown.

## Human tissues and histological analysis

Frozen blocks of human skeletal muscle were obtained from Asterand (Detroit, MI), who acquired appropriate informed consent from patients under the Institutional Review Board (IRB) approval. Subject characteristics are described in *Table 1*.

The sections were fixed in acetone for 10 min, permeabilized with 0.2% (vol/vol) Triton X-100/PBS for 30 min, and then blocked with CAS-Block (Invitrogen) for 10 min at room temperature. Subsequently, the sections were incubated with anti-α-actinin, anti-Rb (1:100, ab24; Abcam) and anti-mDia1 (1:400, ab11173; Abcam) antibodies overnight at 4°C. They were further incubated with appropriate fluorescence-labeled secondary antibodies for 1 hr at room temperature.

## EPS

HSMMs grown on 4-well plates (Nunc, Naperville, IL) were pretreated with IFN-γ and then placed in a C-Dish electrode chamber (IonOptix, Milton, MA) after changing to fresh serum-free media. EPS was applied to HSMMs using a C-Pace pulse generator (IonOptix) at 40 V/60 mm, 1 Hz, 10 ms for 2 days in parallel with TNF-α treatment. The media were changed and TNF-α was repeatedly added every 24 hr. Images of live beating cells were taken using Nikon Eclipse Ti-U microscope, equipped with a 20× objective lens. Images were sequentially acquired with Nikon NIS-Elements imaging software at a frame rate of 15 fps.

**Table 1.** Subject characteristics

| Biosample diagnosis | Normal | Atrophy |
|---|---|---|
| Gender and age, years | Female, 15 | Female, 14 |
| Cancer diagnosis | Osteosarcoma | Synovial sarcoma |
| Cancer location | Shin bone | Soft tissues of shin |
| Height, cm | 166 | 159 |
| Weight, kg | 50 | 50 |
| BMI, kg/m² | 18.14 | 19.78 |

BMI, body mass index

## qPCR

Total RNA extraction, cDNA preparation, and real-time qPCR analyses were performed as described previously (*Kawauchi et al., 2012*). The primer sets used were: *Tk1*, 5′-CATTAACCTGCCCACTGT-3′ forward and 5′-GATCACCAGGCACTTGTA-3′ reverse; *Dhfr*, 5′-TCATGGTTGGTTCGCTAA-3′ forward and 5′-TGAAGAGGTTGTGGTCATT-3′ reverse; *Tfam*, 5′-TGTAGAAGCCACGGTGTT-3′ forward and 5′-ACAACCATCAACTCTGAATAC AAT-3′ reverse; *Ppargc1a*, 5′-TGAAGAGGCAAGA GACAGAATGA-3′ forward and 5′-CACACGCA CACTCCATCAC-3′ reverse; *B2m*, 5′-GCATTCC

TGAAGCTGACA-3′ forward and 5′-CGTGAGTAAACCTGAATCTTT-3′ reverse. After normalization against *B2m*, data show mRNA expression levels relative to control expression levels for each experiment.

## Mass spectrometry analysis

Mass spectrometry analysis was performed by Proteomics International (Perth, Australia). Protein samples were enzymatically digested to produce fragmented peptides and the resulting peptides were analyzed by electrospray ionization mass spectrometry using the Ultimate 3000 nano HPLC system (Dionex, Sunnyvale, CA) in combination with a 4000 Q TRAP mass spectrometer (Applied Biosystems, Foster City, CA). The spectra were analyzed by Mascot sequence matching software (Matrix Science, Boston, MA).

## Acknowledgements

We thank Drs A Bershadsky, D Tenen, M Sheetz, D Shiokawa, H Fujita, Y Shibukawa and S Furuya for their support and advice; Dr S Narumiya for the mDia1 antibody; Dr K Tamai for the Rb-pT821 antibody; and Dr C Koh for the mDia1 plasmid.

## Additional information

### Funding

| Funder | Author |
|---|---|
| Singapore National Research Foundation and the Ministry of Education under the Research Centres of Excellence Programme | Yoichi Taya, Keiko Kawauchi |

The funders had no role in study design, data collection and interpretation, or the decision to submit the work for publication.

### Author contributions

KA, Conception and design, Acquisition of data, Analysis and interpretation of data, Drafting or revising the article; KK, Conception and design, Acquisition of data, Analysis and interpretation of data; HH, YT, Conception and design, Analysis and interpretation of data; MY, Discussed the results, Acquisition of data

### Ethics

Human subjects: The human samples in this study were handled and maintained in accordance with protocols approved by the National University of Singapore-IRB.

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
