## [Decision Letter]

Thank you for sending your work entitled “Cytoplasmic translocation of the retinoblastoma protein disrupts sarcomeric organization” for consideration at *eLife*. Your article has been favorably evaluated by a Senior editor, a Reviewing editor, and 3 reviewers.

The Reviewing editor and the other reviewers discussed their comments before we reached this decision, and the Reviewing editor has assembled the following comments to help you prepare a revised submission. We concur that several important issues need to be addressed during revision:

1) More controls are needed for experiments investigating Rb localization, given the tiny amount of protein that localizes to the cytoplasm in over-expression experiments. Antibody specificity should also be verified, for example in null or knocked down cells.

2) The absence of data on endogenous Rb raises concerns on the real biological significance of the phenomenon described. While addressing this issue may be technically demanding, it is felt that the novelty of this work will be met with skepticism unless the data are really solid and unquestionable.

3) As Rb is known to interact with many proteins, details regarding the mass spec experiments that identified cytoplasmic partners of Rb should be provided, and the reason for selecting specifically mDia1 should be explained.

4) Testing whether TNF-alpha fails to disrupt sarcomeres in Rb null or knocked down myotubes is also an essential control that is currently missing.

5) Other points to be clarified are the pre-treatment with IFN and the very high dose of TNF-alpha. Is the effect observed only with combination of the two molecules and only at this high dose?

6) Data on human muscle biopsies from cancer patients are important but more information should be provided.

---

## [Author Response]

*1) More controls are needed for experiments investigating Rb localization, given the tiny amount of protein that localizes to the cytoplasm in over-expression experiments. Antibody specificity should also be verified, for example in null or knocked down cells*.

In order to corroborate the reliability of our data regarding TNF-α-induced cytoplasmic translocation of Rb, we carried out the nuclear and cytoplasmic fractionation experiments of TNF-α-treated HSMMs in parallel with HSMMs infected with adenoviruses expressing mCherry-HA-Rb or mCherry-HA-NES Rb (new Figure 6). As shown in Figure 6, the vast majority of the exogenous Rb proteins were localized in the nucleus (mCherry-HA-Rb) and the cytoplasm (mCherry-HA-NES Rb). These proteins can therefore be used for the nuclear and cytoplasmic fraction controls (Figure 6, asterisk) and the reliability of our subcellular fractionation experiments are supported as these exogenous Rb proteins properly fractionated into the nuclear and cytoplasmic fractions. We confirmed that in untreated HSMMs, Rb was primarily present in the nucleus and phosphorylated Rb accumulated in the cytoplasm after TNF-α treatment. In addition, we have verified the specificity of Rb antibodies used in immunofluorescence microscopy. The epifluorescence microscopy data clearly shows that the immunofluorescence signal of these antibodies was significantly weakened in Rb knocked down HSMMs (new Figure 5). The information regarding epifluorescence microscopy has been now included in the Materials and methods section.

*2) The absence of data on endogenous Rb raises concerns on the real biological significance of the phenomenon described. While addressing this issue may be technically demanding, it is felt that the novelty of this work will be met with skepticism unless the data are really solid and unquestionable*.

We agree with the reviewers’ concern. Our new data suggests endogenous Rb contributes to TNF-α-induced sarcomeric disorganization because TNF-α failed to induce sarcomeric disorganization in Rb knocked down HSMMs (new Figure 5; see also author response 4). In addition, given that Rb phosphorylation was partially induced by TNF-α treatment and a small population of endogenous Rb was localized in the cytoplasm, we have examined whether TNF-α-induced cytoplasmic Rb is sufficient for the biological function of Rb in the cytoplasm. To address this issue, we have expressed a NES-fused form of Rb at a comparable level to TNF-α-induced cytoplasmic Rb (new Figure 6, asterisk versus the open arrowhead). Under this condition, the repeating pattern of α-actinin distribution was disrupted (Figure 6), which was the same as TNF-α-treated HSMMs (Figure 3). These results suggest that TNF-α-induced phosphorylation of Rb yields sufficient cytoplasmic accumulation of Rb, which may have a pivotal role in TNF-α-induced sarcomeric disorganization.

*3) As Rb is known to interact with many proteins, details regarding the mass spec experiments that identified cytoplasmic partners of Rb should be provided, and the reason for selecting specifically mDia1 should be explained*.

We have now described the information regarding mass spectrometry experiments in greater detail in Figure 7—figure supplement 1 and in the Materials and methods section. The lists of identified proteins are also shown. Sarcomeric disorganization was induced by NES Rb wild-type, but not by NES Rb lacking exon 22. Because the exon 22-deleted mutant Rb is known to be an LXCXE-binding deficient mutant, we hypothesized that the function of cytoplasmic Rb is rendered through its interaction with LXCXE motif-containing proteins. This critical point was therefore addressed, as highlighted by the title of Figure 7. Among the proteins identified (∼27 proteins), we focused on mDia1 (DIAPH1) because only mDia1 contains the LXCXE motif. In addition, “it is noteworthy that the band including mDia1 was hardly detected in mCherry-HA-NES Rb Δexon 22-expressing HSMMs (Figure 7—figure supplement 1, band 1).” This sentence has been now included in the Results section of our revised manuscript because it further supports our idea that mDia1 is a candidate of NES Rb-binding protein and is involved in the function of NES Rb WT.

*4) Testing whether TNF-alpha fails to disrupt sarcomeres in Rb null or knocked down myotubes is also an essential control that is currently missing*.

Following the reviewers’ comment, we have generated a recombinant adenovirus expressing shRNA against human Rb (sh-Rb) and tested whether TNF-α fails to disrupt sarcomeric organization in Rb knocked down HSMMs. Suppression of Rb expression by shRNA was validated by immunoblotting and epifluorescence microscopy (new Figure 5). The sequence for sh-Rb was added to the Materials and methods section. As shown in new Figure 5, TNF-α-induced sarcomeric disorganization was attenuated in sh-Rb-expressing HSMMs when compared to control shRNA-expressing HSMMs. These results support our idea that TNF-α-induced cytoplasmic Rb is important for TNF-α-induced sarcomeric disorganization.

*5) Other points to be clarified are the pre-treatment with IFN and the very high dose of TNF-alpha. Is the effect observed only with combination of the two molecules and only at this high dose*?

We have tested the cells treated with different concentrations of TNF-α (10 or 100 ng/ml) with or without IFN-γ pretreatment. Both phosphorylation of Rb and cytoplasmic translocation of Rb were hardly induced by a high dose of TNF-α (100 ng/ml) alone or a low dose of TNF-α (10 ng/ml) in combination with IFN-γ pretreatment (Figure 12). The amount of TNF-α and IFN-γused in this study would be greater than their circulating levels in cancer patients, but supraphysiological activity of TNF-α is likely to be required in order to obtain a more efficient response in cultured human myotubes. We have now provided relevant discussion in our revised manuscript.Author response image 1.HSMMs were treated with TNF-α (10 or 100 ng/ml) for 2 days with or without IFN-γ pretreatment (100 ng/ml, 8 hr). TNF-α was repeatedly added every 24 hr. **A**. Total cell lysates were prepared and immunoprecipitated with anti-Rb antibody, followed by immunoblotting. The open and solid arrowheads indicate the position of phosphorylated and unphosphorylated Rb, respectively. **B**. Confocal images for Rb and DAPI-stained nuclei. Scale bar, 20 μm.

*6) Data on human muscle biopsies from cancer patients are important but more information should be provided*.

We agree with the reviewers that patient information is important and we have now included this in the Materials and methods section (new Table 1).